# Antimicrobial Effect and Probiotic Potential of Phage Resistant *Lactobacillus plantarum* and its Interactions with Zoonotic Bacterial Pathogens

**DOI:** 10.3390/foods8060194

**Published:** 2019-06-05

**Authors:** Vinod Nagarajan, Mengfei Peng, Zajeba Tabashsum, Serajus Salaheen, Joselyn Padilla, Debabrata Biswas

**Affiliations:** 1Department of Animal and Avian Sciences, University of Maryland, College Park, MD 20742, USA; biovinz@gmail.com (V.N.); murphy7@umd.edu (M.P.); salaheen@umd.edu (S.S.); 2Biological Sciences Program, University of Maryland, College Park, MD 20742, USA; ztabashs@terpmail.umd.edu (Z.T.); joselyn0520@hotmail.com (J.P.); 3Centre for Food Safety and Security Systems, University of Maryland, College Park, MD 20742, USA

**Keywords:** probiotic, *Lactobacillus plantarum*, bacteriophage, zoonotic bacterial pathogen

## Abstract

Development of phage-resistant probiotic particularly *Lactobacillus* is an alternative approach to enhance their beneficial effects as in animal feed supplements. In this study, we developed phage-resistant *Lactobacillus plantarum* (LP*^+PR^*) mutant and compared their antimicrobial effects and probiotic potential against zoonotic bacterial pathogens including *Salmonella enterica* serovar Typhimurium, enterohemorrhagic *Escherichia coli* (EHEC), *Staphylococcus aureus*, and *Listeria monocytogenes* with phage-sensitive *L. plantarum* (LP) strain. LP*^+PR^* strain showed markedly higher growth rate than wild-type LP strain. In co-culture with LP*^+PR^* and in the presence of cell-free cultural supernatants (CFCSs) of LP*^+PR^*, the growth of *S.* Typhimurium, EHEC, *S. aureus*, and *L. monocytogenes* were reduced significantly (*P* < 0.05). The adhesion ability of LP*^+PR^* was slightly higher than the LP on human epithelial INT-407 cells. Most importantly, LP*^+PR^* strain significantly inhibited the adhesive and invasive abilities of all four zoonotic pathogens to INT-407 cells (*P* < 0.05). Moreover, real-time qPCR revealed that in the presence of LP*^+PR^* strain or its CFCSs, expression of virulence genes of these zoonotic bacterial pathogens were suppressed significantly (*P* < 0.05). These findings suggest that the LP*^+PR^* strain is capable of inhibiting major zoonotic bacterial pathogens efficiently and would be a potential candidate for industrial usage in animal production or fermentation.

## 1. Introduction

Lactic acid bacteria (LAB) represents the important group of probiotic due to their ability to exhibit wide-range of health-promoting effects specifically improving gut health and immunity [1]. *Lactobacillus* are most common and widely used probiotic bacteria in the food and dairy industries [2] and the common functional species are *Lactobacillus plantarum* (LP), *L. acidophilus, L. casei, L. delbrueckii, L. fermentum, L. johnsonii, L. paracasei, L. reuteri, L. rhamnosus, and L. salivarius* [3]. Further, benefits of probiotics vastly depend on the metabolites, for example, lactic acid, acetic acid, propionic acid, succinic acid, hydrogen peroxide, carbon dioxide, acetaldehyde, acetoine, reuterin, reutricyclin, and bacteriocins, as well as their ability to produce low molecular weight antimicrobial substances [4,5]. Numerous studies have shown that probiotics help to maintaining the balance of the intestinal microflora, promoting innate and adaptive immune responses, modulating the intestinal immune system [6], detoxifying colonic toxins, lowering serum cholesterol levels [7], promoting lactose tolerance, inhibiting bacterial toxin, producing metabolites essential to the function of intestinal epithelial cells [8], preventing and treating diarrhea, food allergies, acute gastritis, and control of total cholesterol and ratio of LDL and HDL [9].

Commonly LP alone or in combination with other probiotics is used as a starter culture in different fermented foods and beverages as well as feed supplement for farm animal production [10]. LP has proven to possess the ability to grow and survive in various environmental conditions and colonize well in the intestinal tract of humans and other mammals [11,12]. The zoonotic pathogens such as *Salmonella enterica* serovar Typhimurium, enterohaemorrhagic *Escherichia coli* (EHEC), *Staphylococcus aureus*, and *Listeria monocytogenes* can cause diarrhea and gastroenteritis in humans and animals. Thus, LP can block adhesion of enteric pathogens, by attaching themselves on the surface of the intestinal epithelial cells and preventing the entry of those pathogenic bacteria to gastrointestinal epithelium [13,14,15]. In addition, LP showed antioxidant and antimicrobials properties in minced chicken meat during fermentation [16]. Several other studies have also shown that LP could reduce diarrheal infections, abdominal pain and constipation associated with irritable bowel syndrome, bloating, and inhibit the entry of zoonotic enteric pathogens into Caco-2 cells [17,18].

Bacteriophage infections are one of the serious threats to the activity of LAB in the global dairy industry. Particularly, the dairy industry has been accepted the problem with phage contamination in dairy-based foods [19]. The beneficial activities of probiotics can be extremely affected when phage contamination occurs. Especially, phage contamination can lead to significant economic loss, waste of ingredients, low product quality, growth of spoilage and bacterial pathogens, total production loss [20]. In addition, phage populations can also increase if the phage-sensitive probiotic strains are used as a starter culture. Therefore, the development of phage-resistant probiotic strains from phage-sensitive probiotic strains would be excellent alternative strategies to minimize these problems. 

The objective of this study was to develop and characterize the probiotic potential of phage-resistant *L. plantarum* (LP*^+PR^*) strain and their growth inhibitory effect on zoonotic bacterial enteric pathogens, such as *S.* Typhimurium, EHEC, *S. aureus*, and *L. monocytogenes*. In addition, we examined the adherence properties of LP*^+PR^* strain by using INT-407 cell model. We have also investigated the probiotic properties of this LP*^+PR^* in reducing host cell (INT-407) and pathogen interactions and virulence gene expression of zoonotic pathogens in the presence of LP*^+PR^* strain by quantitative real-time PCR (RT-qPCR).

## 2. Materials and Methods

### 2.1. Bacterial Strains and Growth Conditions

A bacteriophage sensitive *Lactobacillus plantarum* (LP, ATCC39542), and four enteric bacterial pathogens including *Salmonella enterica* serovar Typhimurium (ATCC14028), enterohemorrhagic *Escherichia coli* EDL933 (EHEC, ATCC700927), *Staphylococcus aureus* (ATCC29740) and *Listeria monocytogenes* (ATCC19115) were used in this study. In addition, phage-resistant mutant strain was generated from phage sensitive LP and named as LP*^+PR^*. Both LP and LP*^+PR^* were grown on de Man Rogosa Sharpe (MRS) agar or broth overnight at 37 °C under aerobic condition with 5% CO_2_ (Thermo Fisher Scientific, MA, USA). The bacterial pathogens *S.* Typhimurium, EHEC, *S. aureus*, and *L. monocytogenes* were grown on Luria-Bertani (LB) agar/broth, MacConkey agar, Tryptic Soy agar (TSA), and Brain Heart Infusion (BHI) agar (EMD Chemicals Inc., NJ, USA), respectively, and incubated overnight at 37 °C under aerobic conditions (Thermo Fisher Scientific, MA, USA).

### 2.2. Bacteriophage Culture and Propagation

The commercial J1 bacteriophage (ATCC 27139-B1) was purchased from American Type Culture Collection, suspended in saline magnesium buffer (SM buffer) and stored at 4 °C. J1 bacteriophage was propagated in MRS broth containing host strain LP at 37 °C under aerobic condition with 5% CO_2_ for 48 h. The culture was centrifuged at 3000× *g* for 20 min and filtered through a 0.2 μm sterilized filter. The bacteriophages were further purified following method previously described [21].

### 2.3. Mammalian Cell and Culture Conditions

Human epithelium cell line (INT-407 CCL-6) was purchased from American Type Culture Collection and cultured according to the method described previously with slight modifications [22]. Briefly, cells were routinely cultured in Dulbecco’s modified Eagle medium (DMEM) (HyClone Laboratories Inc., UT, USA) supplemented with 10% fetal bovine serum (FBS) and 50 µg/mL gentamycin. Cells were grown at 37 °C in 24-well tissue culture plates (BD Falcon, NJ, USA) in a 5% CO_2_-95% humidity atmosphere. The post-confluent INT-407 epithelial cell monolayers were washed three times with sterile PBS and cell culture medium was replaced by fresh DMEM every 48 h. 

### 2.4. Isolation of Phage-Resistant L. plantarum (LP^+PR^) Mutant

To isolate *LP^+PR^* mutant strain, the agar plate method was used as described previously with some modifications [23]. An overnight culture of LP in MRS supplemented with 10 mmol/L CaCl_2_ was infected with different ration of J1 bacteriophage (Multiplicity of infection: 1, 0.1, and 0.01). An aliquot of 0.5 mL overnight liquid culture of sensitive strain was inoculated in 4.5 mL 0.5% MRS soft agar with 10 mmol/L CaCl_2_. Pre-warmed MRS agar plate was overlaid with the 5 mL mixture mentioned above. After the soft agar solidified, 25 µL of the J1 phage suspension was spotted on the soft agar, followed by incubation at 37 °C with 5% CO_2_ for 48 h. After incubation, survived colony of LP was isolated and cultured on fresh MRS agar plate for three more consecutive phage-resistant selection described above. The isolated colony after three consecutive selection was isolated, cultured in MRS broth, and stored as presumptive LP*^+PR^* strain.

To confirm LP*^+PR^* strain, the genomic DNA of J1 bacteriophage, LP and LP*^+PR^* strains were isolated using DNeasy Blood & Tissue Kit (Qiagen, CA, USA). Gene encoding for phage tail protein (*ptp*) in J1 bacteriophage was targeted for LP*^+PR^* confirmation. PCR was performed in Mastercycler (Eppendorf, NY, USA) and programmed for initial 2 min 94 °C followed by 40 cycles of 94 °C for 1 min, 45 °C for 30 s, and 72 °C for 2 min, followed by a final elongation of 10 min at 72 °C. Primers were synthesized from Eurofins MWG Operon LLC (Louisville, KY, USA). PCR products were resolved by electrophoresis through 1.5% agarose (Sigma-Aldrich) gel at 120 V for 45 min in Tris-acetate-EDTA (TAE) buffer. 

### 2.5. Growth and Survival of LP and LP^+PR^ Strains

LP and LP*^+PR^* strains were grown overnight at 37 °C in MRS broth. The overnight cultured bacterial strains were centrifuged, washed three times with PBS and adjusted to concentration at 1 × 10^6^ CFU/mL. Both bacterial suspensions were diluted to 1:10 into fresh MRS broth and incubated at 37 °C. Samples were taken at 0, 6, 12, 24, 48, and 72 h intervals, serially-diluted, and plated on MRS agar. Plates were incubated under micro-aerophilic condition at 37 °C for 48 h and surviving bacteria were counted as colony-forming units per milliliter (CFU/mL). 

### 2.6. Co-Culture of LP or LP^+PR^ Strain with Zoonotic Bacterial Pathogens

LP, LP*^+PR^*, *S.* Typhimurium, EHEC, *S. aureus*, and *L. monocytogenes* were grown overnight at 37 °C as described above. Aliquot of 400 µL LP or LP*^+PR^* strains (1 × 10^8^ CFU/mL) was co-cultured with 400 µL of enteric pathogens (1 × 10^5^ CFU/mL) in 3.2 mL of DMEM supplemented with 5% heat-inactivated FBS. Cultures were incubated at 37 °C under aerobic condition and samples were taken at 0, 6, 24, and 48 h time points. Bacterial counts were determined by plating of 10-fold serial dilutions and spread on specific agars. 

### 2.7. Cell-Free Cultural Supernatants (CFCSs) of LP and LP^+PR^ Strains on Growth of Zoonotic Pathogens

Overnight (18 h) liquid cultures of LP and LP*^+PR^* strains were grown in DMEM supplemented with 5% FBS and centrifuged at 4000× *g* for 20 min (Thermo fisher Scientific Inc., MA, USA). Cells were washed twice with PBS and CFCSs were further filtered by sterile syringe 0.2 µm filter (VWR Inc, PA, USA) and stored at 4 °C [24,25]. The bacterial cell suspension containing 400 µL (1 × 10^5^ CFU/mL) of *S.* Typhimurium, EHEC, *S. aureus*, or *L. monocytogenes* and 400 µL CFCSs collected from LP or LP*^+PR^* strain were inoculated in a separate culture tubes with 3.2 mL of DMEM and 5% FBS. The culture tubes were incubated at 37 °C and samples were plated on specific agar plates at 0, 6, 24, and 48 h. Plates were incubated at 37 °C for 24 h and surviving bacteria were counted as CFU/mL. 

### 2.8. Host Cells-LP or LP^+PR^ Interactions

The adherence activity of LP or LP^+PR^ mutant strains were performed as described previously with some modifications [26]. The adherence abilities of LP and LP*^+PR^* strains were examined using INT-407 cells. For adhesion assay, the INT-407 cells were seeded at a concentration of 2 × 10^5^ cells/well in 24-well plate with 900 µL DMEM containing 10% FBS. The bacterial cells were cultured at 37 °C for overnight, centrifuged, and washed three times with sterile PBS before the adhesion assay. 100 µL of LP or LP*^+PR^* strains (1 × 10^8^ CFU/mL) suspended in 1 mL DMEM medium with 10% FBS were added to wells and then plates were incubated at 37 °C in 5% CO_2_-95% air for 2 h. After incubation, the unattached cells were washed three times with sterile PBS (pH 7.4) and cells were lysed by addition of 0.1% Triton X-100 for 15 min. The adherence level of LP and LP*^+PR^* strains was measured by plating serial dilutions adhered bacteria on MRS agar. The plates were incubated at 37 °C for 48 h. 

### 2.9. Inhibition of Adherence and Invasion of Zoonotic Enteric Pathogens with CFCSs Collected from LP and LP^+PR^ Strains

In order to determine cell adhesion and invasion of *S.* Typhimurium, EHEC, *S. aureus*, and *L. monocytogenes* by co-culture and CFCSs of LP or LP*^+PR^* strains was carried out following the method described previously [22]. The INT-407 cells grown in 24-well plate with 800 µL DMEM containing 10% FBS were pre-treated with 100 µL DMEM (control), 1 × 10^8^ CFU/mL of LP or LP*^+PR^*, CFCSs of LP or LP*^+PR^* incubated at 37 °C for 1 h. After pre-treatment, INT-407 cells were washed three times with sterile PBS and fresh DMEM with 10% FBS was added. A 100 µL of enteric pathogenic bacteria (1 × 10^5^ CFU/mL) were then added to the well. After an additional 2 h incubation, the unattached cells were washed three times with sterile PBS. The cells were lyzed by 0.1% Triton X-100 for 15 min, serially diluted, and plated on specific agar plates for enumerating the adhesive bacterial cells. Cell invasive activity was measured by further 1 h incubation of washed monolayers in DMEM containing 10% FBS and 100 µg/mL gentamycin. The cells were washed three times with PBS and lysed with Triton X-100, serially diluted, and plated on specific agars. 

### 2.10. RNA Extraction and cDNA Synthesis

All four zoonotic pathogenic bacteria were grown overnight in the presence and absence of CFCSs of LP or LP*^+PR^* strains, and RNA was extracted according to the methods of Peng et al. (2015) [22]. Bacterial cells were washed three times with ice-cold PBS and 1 mL TRIzol reagent (Life Technologies Co., Carlsbad, CA, USA) was added to lyse the cells for 5 min at room temperature. After the cells were lysed, 200 µL of chloroform was added and samples were mixed and kept at room temperature for 3 min. The samples were centrifuged at 13,000× *g* for 15 min at 4 °C. After centrifugation, the supernatant was collected and mixed with 500 µL of isopropanol and allowed to stand for 10 min at room temperature before being centrifuged at 13,000× *g* for 15 min at 4 °C. The supernatant was removed, washed with 1 mL of 75% ethanol and centrifuged at 7000× *g* for 5 min at 4 °C. The remaining supernatant was removed, and the pellets were placed in bio-safety cabinet for 10 min to dry. The RNA pellet was fully dissolved in 50 µL RNase-free water and quantified by a NanoDrop spectrophotometer (Thermo Fisher Scientific).Using a qScript cDNA SuperMix protocol (Quanta Biosciences, MD, USA), one µg of total RNA was mixed with qScript cDNA SuperMix containing optimized concentration of MgCl_2_, dNTPs, random primers, qScript reverse transcriptase, and RNase inhibitor protein. Subsequently, the reaction mixture was incubated at 25 °C for 5 min, 42 °C for 30 min, and 85 °C for 5 min to terminate the reaction. Finally, the synthesized cDNA was stored at −20 °C until further use. 

### 2.11. Quantitative RT-PCR Assay

The PCR reaction mixture containing 10 µL of PerfeCTa STBR Green FastMix, 1 µL of forward and reverse primers (100 nM), 2 µL of cDNA (10 ng), and 4 µL of RNase-free water were amplified using an Eco Real-Time PCR system (Illumine, CA, USA). The reaction conditions were as follows: 95 °C for 30 s, followed by 40 cycles at 95 °C for 5 s, 55 °C for 15 s, and 72 °C for 10 s. All the virulence genes were estimated by comparative fold change. The CT values of genes were normalized to the housekeeping gene 16s rRNA and the relative expression levels of target genes were compared between control and treatment. The fold change in terms of expression of each individual virulence gene was calculated as ΔΔCT = (CT(target mRNA) − CT(reference mRNA))treatment − (CT(target mRNA) − CT(reference mRNA))control [27].

### 2.12. Statistical Analysis

All statistical analyses were performed with one-way ANOVA using SPSS software (version 20.0) and data were expressed as mean ± standard deviation. Data were analyzed at the significance level of *P* < 0.05. 

## 3. Results

### 3.1. Confirmation of Bacteriophage Resistant LP^+PR^ Strain

To confirm the bacteriophage resistance characteristics of LP*^+PR^* strain, we targeted J1 bacteriophage *ptp* gene in LP*^+PR^* strain genome. Genomic DNA from wild-type LP and J1 bacteriophage containing LP*^+PR^* strain as well as J1 bacteriophage were amplified and visualized in the gel electrophoresis image. We found that LP*^+PR^* genome contained the 300 bp of *ptp* gene of J1 bacteriophage (Appendix A). These results revealed that phage-resistant mutant strain was successfully developed from the phage-sensitive wild-type LP strain by J1 bacteriophage. 

### 3.2. Comparative Growth and Survival Ability of LP and LP^+PR^ Strains

The growth and survival ability of LP and LP*^+PR^* strains were compared by enumerating viable cell counts after 0, 6, 12, 24, 48, and 72 h time intervals and shown in Figure 1. LP*^+PR^* strain showed greater survival rate than the wild-type LP strain after 12 h incubation. Most importantly, the bacteriophage resistant LP*^+PR^* strain (10.46 log CFU/mL) exhibited significantly higher survival rate in comparison to wild-type LP strain (9.41 log CFU/mL) at 24 h (*P* < 0.05). These results demonstrate that LP and LP*^+PR^* strain was able to grow and survive under in vitro condition but LP*^+PR^* could grow and survive better than the wild-type LP strain. 

### 3.3. Competitive Inhibition of Zoonotic Pathogens in Co-Culture Condition

The growth of *S.* Typhimurium, EHEC, *S. aureus*, and *L. monocytogenes* in the presence or absence of LP or LP*^+PR^* strains was monitored at 37 °C. Growth inhibition of all four enteric pathogens are presented in Figure 2. In this study, we found that both LP and LP*^+PR^* strains significantly inhibited the growth of bacterial enteric pathogens after 24 h of incubation. Moreover, the number of viable EHEC and *S. aureus* in co-culture with either LP or LP*^+PR^* strain decreased significantly after 6 h of incubation (*P* < 0.05). We observed that wild-type LP and mutant LP*^+PR^* significantly inhibited the growth of *S.* Typhimurium by 2.88 and 3.21 log CFU/mL, EHEC by 3.38 and 3.22 log CFU/mL, *S. aureus* by 4.63 and 5.16 log CFU/mL and *L. monocytogenes* by 3.86 and 4.68 log CFU/mL, respectively at 48 h.

### 3.4. Effects of CFCSs of LP and LP^+PR^ on Growth of Zoonotic Pathogens

The either CFCSs collected from LP or LP*^+PR^* showed growth inhibitory effects on *S.* Typhimurium, EHEC, *S. aureus*, and *L. monocytogenes* (Figure 3). Both CFCSs collected from LP or LP*^+PR^* showed tendency to inhibit the growth of EHEC at 6 h. After 24 h, CFCS collected from LP*^+PR^* significantly reduced the growth of *S.* Typhimurium (1.93 log CFU/mL), EHEC (2.99 log CFU/mL), *S. aureus* (2.58 log CFU/mL), and *L. monocytogenes* (2.93 log CFU/mL). Compared to the control (only cell culture medium with *S. aureus*) and CFCS collected from wild-type LP, CFCSs of the LP*^+PR^* significantly inhibited the growth of *S. aureus* after 48 h (Figure 3C). CFCSs collected from wild-type LP at 48 h time point significantly reduced growth of *S.* Typhimurium, EHEC, *S. aureus,* and *L. monocytogenes* 2.91 log CFU/mL, 3.23 log CFU/mL, 2.70 log CFU/mL, and 4.41 log CFU/mL, respectively (Figure 3) compared to the control. In the same study, CFCSs collected from LP*^+PR^* showed more intensive reduction of growth of *S.* Typhimurium (3.53 log CFU/mL), EHEC (3.12 log CFU/mL), *S. aureus* (3.95 log CFU/mL), and *L. monocytogenes* (4.43 log CFU/mL) at 48 h time point. 

### 3.5. Adhesion Abilities of LP and LP^+PR^ to Mammalian Cells

Adhesion of LP*^+PR^* strain to host cells was examined using human epithelial INT-407 cells (Figure 4). Adhesion abilities of wild-type LP strain were recorded as 6.29 log CFU/mL, 6.77 log CFU/mL, 6.73 log CFU/mL at 2, 4, and 24 h, respectively. Whereas the mutant LP*^+PR^* adhered aggressively to the INT-407 cells at 6.39 log CFU/mL, 6.87 log CFU/mL, and 6.79 log CFU/mL at 2, 4, and 24 h, respectively. We found that LP*^+PR^* strain can adhere more effectively to INT-407 cells compare to wild-type LP strain. However, there is no significant difference between wild-type LP and LP*^+PR^* strain.

### 3.6. Role of LP and LP^+PR^ in Alteration of Pathogen-Host Cell Interactions

Pre-treatment of INT-407 cells with LP or LP*^+PR^* or CFCSs collected from either wild-type LP or mutant LP*^+PR^* strain, reduced both adherence and invasion abilities of *S.* Typhimurium, EHEC, *S. aureus*, and *L. monocytogenes* (Figure 5; *P* < 0.05). It was observed that wild-type LP or mutant LP*^+PR^* strain reduced adhesion abilities of *S.* Typhimurium by 0.81 and 0.88 log CFU/mL, EHEC by 0.18 and 0.32 log CFU/mL, *S. aureus* by 0.93 and 0.85 log CFU/mL, and *L. monocytogenes* by 0.45 and 0.44 log CFU/mL, respectively (Figure 5A). In the presence of CFCSs collected from wild-type LP or mutant LP*^+PR^* significantly inhibited adherence activity of 0.41 and 0.43 log CFU/mL (*S.* Typhimurium), 0.50 and 0.54 log CFU/mL (EHEC), 0.56 and 0.87 log CFU/mL (*S. aureus*), and 0.83 and 0.89 log CFU/mL (*L. monocytogenes*), respectively (Figure 5C). 

In the same assay, we found that pre-treatment with wild-type LP or mutant LP*^+PR^* strain significantly reduced invasion abilities of all four pathogens (0.28 and 0.52 log CFU/mL for *S.* Typhimurium, 0.27 and 0.41 log CFU/mL for EHEC, 0.51 and 0.73 log CFU/mL for *S. aureus*, and 0.36 and 0.44 log CFU/mL for *L. monocytogenes*) at various levels compared to control (Figure 5B). Meanwhile, the invasiveness of *L. monocytogenes* in the presence of mutant LP*^+PR^* strain was significantly reduced compared to wild-type LP strain (Figure 5B; *P* < 0.05). We found that invasion abilities of *S.* Typhimurium, EHEC, *S. aureus*, and *L. monocytogenes* were reduced significantly in the presence of either CFCSs collected from wild-type LP or LP*^+PR^* strains or live probiotics LP or LP*^+PR^* at range from 0.5 log CFU/mL to 1 log CFU/mL (*P* < 0.05) (Figure 5). 

### 3.7. Alteration of Virulence Gene Expression of Zoonotic Pathogens in Presence of CFCSs Collected from LP and LP^+PR^

The analysis of virulence gene expression of *S.* Typhimurium, EHEC, *S. aureus*, and *L. monocytogenes* in the presence of CFCSs collected from either wild-type LP or mutant LP*^+PR^* strains is presented in Figure 6. In comparison with control, all four pathogens treated with CFCSs collected from either wild-type LP or mutant LP*^+PR^* altered the expression of tested virulence genes of all four pathogens. In the presence of LP and LP*^+PR^* CFCSs, the relative expression levels of *fliC*, *fliD, motB, nmpC, hilA, hilD*, and *mig5* were significantly downregulated in *S.* Typhimurium. Moreover, a significant downregulation of *mig5* was found in LP*^+PR^* when compared to control and wild-type LP strain (*P <* 0.05). For EHEC (Figure 6B), the virulence genes *eaeA, fliC, tir, espA*, and *espD* were altered in the presence of CFCSs of wild-type LP or LP*^+PR^*, though no statistically significant differences were found in the relative expression of *tir, espA, espD*, and *fliC*. The virulence genes levels of *norA, norB, norC, sdrM, sepA,* and *mdeA* were altered in *S. aureus* when treated with CFCSs collected from either wild-type LP or LP*^+PR^* strains, specifically with a significant fold-change in *norB, norC, sdrM, sepA,* and *mdeA* compared to control (*P <* 0.05). Finally, in the presence of CFCSs of wild-type LP or LP*^+PR^*, a significant down regulation of *hlyA, iap, fbp, motA,* and *motB* expressions was found in *L. monocytogenes* (*P <* 0.05). 

## 4. Discussion

LAB especially LP have the great potential to produce antimicrobial substances which can inhibit the growth of zoonotic bacterial pathogens and prevent them from infections [28]. To provide such benefits to the animal host, probiotics should have properties like ability to adhere and colonize to intestinal cells, produce bio-active metabolites, prevent infection from pathogenic bacteria, survive in acidic environment, and induce immunomodulatory properties. All these beneficial effects could be demolish in the presence specific bacteriophage of the probiotic in the same ecosystem if the introduced probiotic is sensitive to bacteriophage. 

To overcome this potential risk, isolation of phage-resistant mutants from phage-sensitive probiotic strains is a convenient, simple and natural approach that has no regulatory restrictions. In the present study, we isolated mutant LP*^+PR^* strain by a soft-agar overlay method and confirmed by bacteriophage gene *ptp* in the genome of LP*^+PR^* strain. The LP*^+PR^* isolated in this study showed significantly higher survival rate than the phage-sensitive LP strain at 24 h. These results revealed that the LP*^+PR^* isolated in this work exhibited phage gene in the genome and higher growth and survived longer than the phage-sensitive LP strain. 

The adhesive ability of LAB is considered as an important feature for their use of probiotics and it allows the probiotics to exert their benefits on animal host [5,10]. In vivo bacterial adherence study was difficult to evaluate due to differentiate from nonadherent bacteria in hosts. Therefore, in vitro cell line model was commonly used to evaluate the adherence level of bacteria. In the present study, we used epithelial INT-407 cells for adherence level of LP*^+PR^* strain and compared the adherence level with wild-type LP strain. There were no significant differences amongst adhesion of LP and LP*^+PR^* strains to INT-407 cells. However, LP*^+PR^* strain showed numerically higher adhesion than the wild-type LP strain (Figure 4). Our findings demonstrated that LP*^+PR^* strain could be able to successfully adhere to intestinal epithelial cells. Similar results were reported that phage-resistant *L. delbruekii* can adhere to colon adenocarcinoma Caco2 intestinal cells [26]. In the same study, they also showed phage-resistant mutant inhibited the adhesion and invasion abilities of the zoonotic enteric pathogen, *S.* Enteritidis to Caco2 cells. The initial step for zoonotic infection is crossing the intestinal epithelial cells in hosts. Reducing enteric pathogen adhesion and invasion to the intestinal epithelium and altering the virulence genes can significantly prevent zoonotic cross-contamination [26].

In the present study, we observed that CFCSs of LP*^+PR^* strain reduced the cell attachment and invasive abilities of *S.* Typhimurium, EHEC, *S. aureus*, and *L. monocytogenes* on epithelial INT-407 cells (Figure 5). In addition, adhesion of probiotic bacteria in the gastrointestinal tract stimulating host gut immune responses [29]. Oelschlaeger et al. (2010) [30] demonstrated that production of antimicrobial compounds produced by probiotic bacterial strains contributes to inhibiting on attachment of the pathogenic bacteria. Numerous studies have also shown that organic acids or bacteriocin produced by *Lactobacillus* may also help in preventing the growth of bacterial enteric pathogens in the animals gut [31,32]. Moreover, another study by Das et al. (2013) [33] reported that CFCS collected from LP inhibited the growth and invasion of *Salmonella* spp. in HCT-116 colon cells.

Four zoonotic enteric bacterial pathogens were investigated for their virulence gene expression in the presence of CFCSs collected from LP or LP*^+PR^* strain. Based on our results, we found that all the targeted virulence genes were significantly altered the expression by CFCSs of LP or LP*^+PR^*. The virulence gene of flagella has been well studied in *S.* Typhimurium [34]. Flagella can play an important role in colonization by providing motility to target cells, and receptors and motility genes were shown to be essential for invade into cultured epithelial cells [35]. In this study, virulence genes of *S.* Typhimurium *fliC, fliD, motB, nmpC, hilA, hilD*, and *mig5* were found to be downregulated by LP*^+PR^*. Therefore, alteration of these genes could lead to reduced attachment and invasive abilities of *S.* Typhimurium to the host cells. The locus of enterocyte effacement Pathogenicity Island which encodes two genes, *eae* (intimin) and *tir* (translocated intimin receptor), has been reported to play important role in EHEC pathogenesis and form attaching and effacing in various in vitro cell culture models [36]. Downregulation of EHEC intimin virulence gene in the presence of CFCSs collected from LP*^+PR^* provide us an explanation for the reduced cell adhesion and invasion to the host cells. Multi-drug resistance (MDR) efflux pump genes in *S. aureus* have been well studied and are responsible for resistance to multiple antibiotics especially methicillin [37,38,39,40].

Moreover, these MDR efflux pump genes are involved in an increased virulence activity in *S. aureus* to the host’s gastrointestinal tract. In order to assess the roles of CFCSs of LP*^+PR^* against *S. aureus*, we selected MDR efflux pump genes *norA, norB, norC, sdrM*, *sepA,* as well as methicillin resistance mediatory gene *mecA* in the presence of CFCSs from LP*^+PR^* strain. A significant alteration of MDR efflux pump genes and *mecA* gene were detected when the *S. aureus* was pretreated with CFCSs of LP or LP*^+PR^*. Fournier and Hooper, (2000) [41] investigated MDR efflux genes and found that they can be involved in *S. aureus* adhesion. Flagella and motility related genes from *L. monocytogenes* are essential for initial cell surface attachment and cell invasion activities [42,43]. Therefore, we tested the *L. monocytogenes* virulence genes such as *hlyA*, *iap, fbp, motA,* and *motB* in the presence of CFCSs of LP*^+PR^*. Remarkably, the virulence gene expression was significantly downregulated by LP*^+PR^* strain. These virulence genes results are correlated with cell adhesion and invasion abilities of *S.* Typhimurium, EHEC, *S. aureus*, and *L. monocytogenes* to INT-407 cells. The results from our study clearly demonstrate that mutant LP*^+PR^* strain showed similar or better probiotic properties than wild-type LP strain.

In summary, the present work demonstrated that the bacteriophage resistance LP*^+PR^* strain can grow faster than wild-type LP strain and inhibited the growth and survival of *S.* Typhimurium, EHEC, *S. aureus*, and *L. monocytogenes*. Based on the in vitro results, LP*^+PR^* shows promising probiotic functional properties. Specifically, LP*^+PR^* was able to adhere to epithelial cells and reduced the adherence and invasion abilities of *S.* Typhimurium, EHEC, *S. aureus*, and *L. monocytogenes* to INT-407 cells. Thus, the LP*^+PR^* strain could be a potential candidate for further in vivo studies to development of potential probiotic strain in feed supplement of farm animal production.

## Figures and Tables

**Figure 1 foods-08-00194-f001:**
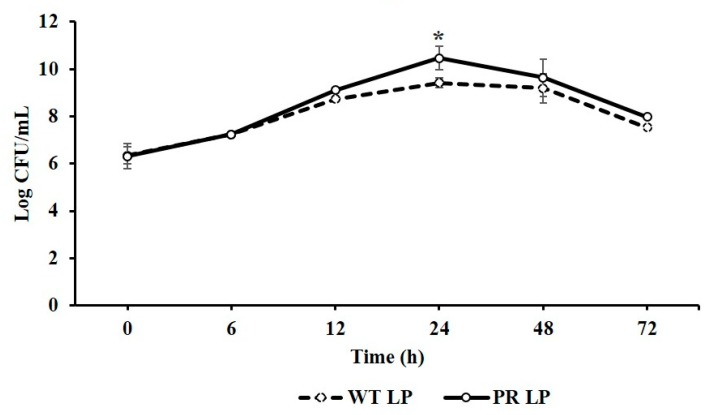
Growth pattern of wild-type *Lactobacillus plantarum* (LP) and phage-resistant *Lactobacillus plantarum* (LP*^+PR^*) strains at 0, 6, 12, 24, 48, and 72 h. Error bars indicate standard deviation from 6 parallel trials. Asterisks indicate the significant growth when compared to wild-type LP strain at *P* < 0.05.

**Figure 2 foods-08-00194-f002:**
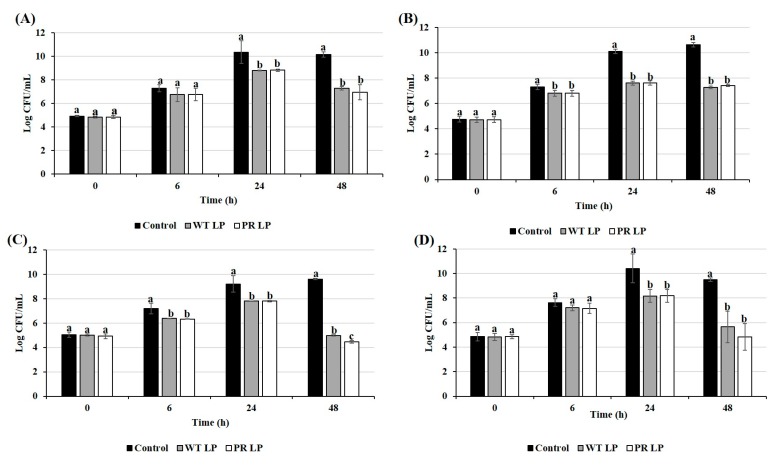
Growth inhibition of *S.* Typhimurium (**A**), enterohemorrhagic *Escherichia coli* (EHEC) (**B**), *S. aureus* (**C**), and *L. monocytogenes* (**D**) in co-culture of wild-type LP and LP*^+PR^* strains at 0, 6, 24, and 48 h. Error bars indicate standard deviation from 6 parallel trials. Different letters (a–c) at each time point indicate the significant growth reduction when compared to control at *P* < 0.05.

**Figure 3 foods-08-00194-f003:**
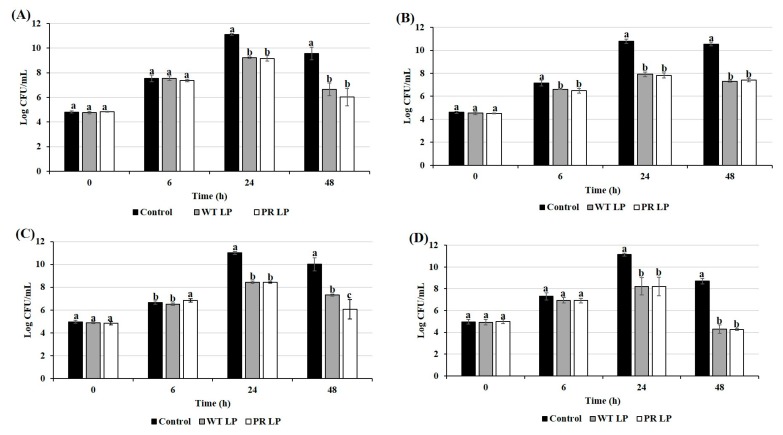
Growth inhibitory effects of cell-free cultural supernatants (CFCSs) from wild-type LP and LP*^+PR^* strains on *S.* Typhimurium (**A**), EHEC (**B**), *S. aureus* (**C**), and *L. monocytogenes* (**D**) at 0, 6, 24, and 48 h. Error bars indicate standard deviation from 6 parallel trials. Different letters (a–c) at each time point indicate the significant growth reduction when compared to control at *P* < 0.05.

**Figure 4 foods-08-00194-f004:**
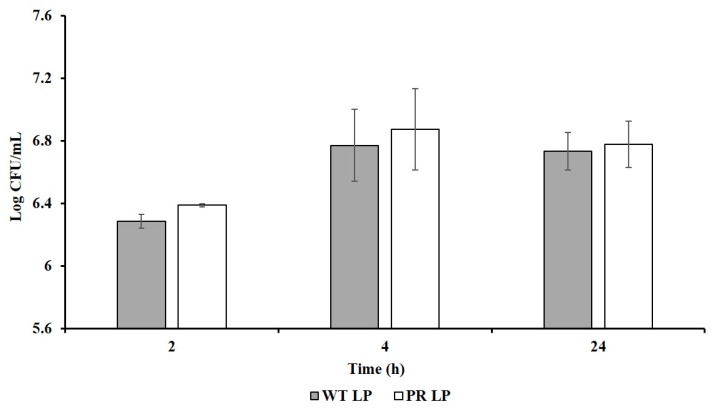
Adherence ability of wild-type LP and LP*^+PR^* strains to INT-407 cells. Error bars indicate standard deviation from 6 parallel trials.

**Figure 5 foods-08-00194-f005:**
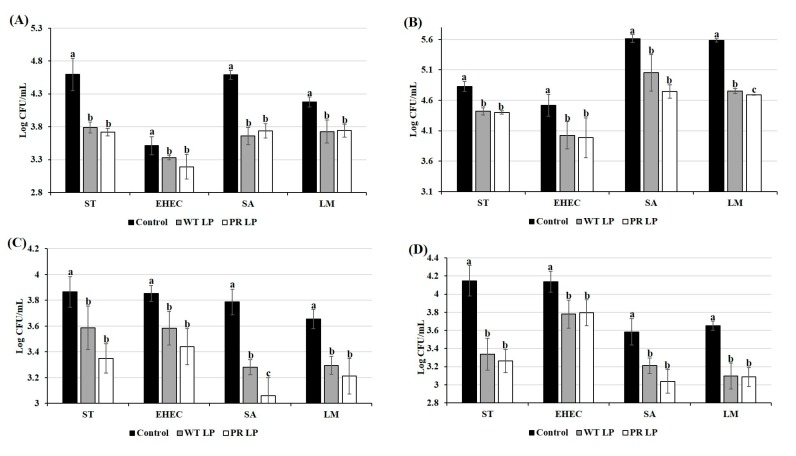
Cell adhesion (**A**,**C**) and invasion (**B**,**D**) abilities of *S.* Typhimurium, EHEC, *S. aureus*, and *L. monocytogenes* to INT-407 cells with pre-treatment of wild-type LP and LP*^+PR^* strains (**A**,**B**) and the CFCSs collected from LP and LP*^+PR^* (**C**,**D**). Error bars indicate standard deviation from 6 parallel trials. Different letters (a–c) are significantly different when compared to control at *P* < 0.05.

**Figure 6 foods-08-00194-f006:**
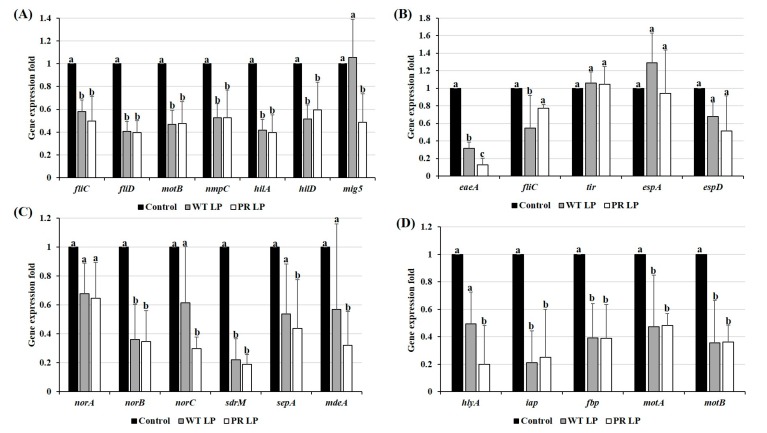
Relative expression of virulence genes of *S.* Typhimurium (**A**), EHEC (**B**), *S. aureus* (**C**), and *L. monocytogenes* (**D**) treated with CFCSs of wild-type LP and LP*^+PR^* strains. Error bars indicate standard deviation from 6 parallel trials. Different letters (a–c) are significantly different at *P* < 0.05.

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
