# Peer review of "Antimicrobial Effect and Probiotic Potential of Phage Resistant Lactobacillus plantarum and its Interactions with Zoonotic Bacterial Pathogens"

_foods, 2019, doi:10.3390/foods8060194_

Reviewer 1 Report

In my opinion the study is well designed and presented. On the other hand, the paper has scientific value. I advise only a few  minor revisions.

Line 15 and title: “probiotic potential against zoonotic bacterial pathogens…” Is probiotic potential only the activity against pathogens? With this point of view, title of the paper also should be re-considered. Antimicrobial effect of this probiotic strain can be highlighted instead of correlating probiotic potential with -only- antimicrobial effect.

Line 19, 76 and whole manuscript: S. Typhimurium is not italic and “t” shouldn’t be capital

Line 204: under the topic after this line, there is “or” between two cfu results… Instead of using “or” it is better to use “respectively” when expressing two different results

Author Response

Reviewer #1:

In my opinion the study is well designed and presented. On the other hand, the paper has scientific value. I advise only a few minor revisions.

1. Line 15 and title: “probiotic potential against zoonotic bacterial pathogens…” Is probiotic potential only the activity against pathogens? With this point of view, title of the paper also should be re-considered. Antimicrobial effect of this probiotic strain can be highlighted instead of correlating probiotic potential with -only- antimicrobial effect.

            We thank the reviewer for the insightful comment. As per the reviewer suggestion, we have modified the title as “Antimicrobial Effect and Probiotic Potential of Phage Resistant Lactobacillus plantarum and its Interactions with Zoonotic Bacterial Pathogens” and also revised the abstract section (Line #15).

2. Line 19, 76 and whole manuscript: S. Typhimurium is not italic and “t” shouldn’t be capital

            We value the reviewer concern. In older literature, “Salmonella typhimurium” is used but in recent literature the serovar name is conventionally capitalized and not italicized. We have been following the same style in our previous research articles. Here with some reference articles for your kind perusal:

1.      Peng et al. Metabolites produced during the growth of probiotics in cocoa supplementation and the limited role of cocoa in host-enteric bacterial pathogen interactions. Food control 2015, 53:124-133.

2.      Peng et al. Lactobacillus casei and its byproducts alter the virulence factors of foodborne bacterial pathogens. Funct. Foods 2015, 15, 418–428.

3. Line 204: under the topic after this line, there is “or” between two cfu results… Instead of using “or” it is better to use “respectively” when expressing two different results

            Again we appreciate the reviewer’s valuable comment. Based on the reviewer’s suggestion, we have revised the results section (line #204) in the manuscript.

Reviewer 2 Report

The present study is an interesting work though authors are needed to proceed to implementations in order to well-describe their study and improve the overall quality of their research.

Here they are manuscript recommendations required prior to publication.

Authors could enforce introduction, i.e why is of importance to study phage registrant potential?

Authors could look for more recent literature in general.

Line 26: pls specify ‘’ in animal production or fermentation’’ what do you mean by that?

This research refers to zoonotic pathogens however there is no such infos in literature except for human enteropathogens.

Lines 16-18: any recent ref?

In section 2.4 and 2.8 references should be provided

Section: 2.10 briefly describe the extraction method, did you use any phenol? Did you proceed to dnase treatment, measure RNA purity in some way?

Could you specify which method did you use for gene expression, Livak? Pfaffl? Cts were logged out? What is the expression ‘threshold’ of expression change?

Discussion section: Not adequate with other studies outcomes.  Most of the literature provided is too old. If no comparison is feasible with other or current literature should be emphasized as it would be the novelty of the study.

Author Response

Reviewer #2:

The present study is an interesting work though authors are needed to proceed to implementations in order to well-describe their study and improve the overall quality of their research.

Here they are manuscript recommendations required prior to publication.

1. Authors could enforce introduction, i.e why is of importance to study phage registrant potential?

            Thank you for the valuable suggestions. We acknowledge the reviewer advice and revised our Introduction section in the manuscript.

2. Authors could look for more recent literature in general.

            Based on the suggestions from the reviewer, we have deleted older literature and added recent literature in the revised manuscript.      

3. Line 26: pls specify ‘’ in animal production or fermentation’’ what do you mean by that?

We appreciate the reviewer’s effort for pointing out this. Probiotics are considered as growth and health stimulators and are used extensively in animal feeding, especially in pig and poultry production (Simon et al. Probiotic feed additives - effectiveness and expected modes of action, J. Anim. Feed Sci. 2001, 10, 51-67 DOI: 10.22358/jafs/70012/2001). Moreover, phage infections are a significant factor in the dairy industry and also the primary cause of fermentation failure in the milk transformation industry (1. Deng et al. Phenotypic, fermentation characterization, and resistance mechanism analysis of bacteriophage-resistant mutants of Lactobacillus delbrueckii ssp. bulgaricus isolated from traditional Chinese dairy products, J. Dairy Sci. 2018, 101, 1901-1914 DOI:  10.3168/jds.2017-13823; 2. Garneau et al. Bacteriophages of lactic acid bacteria and their impact on milk fermentations, Microb. Cell Fact. 2011, 10, S20 DOI: 10.1186/1475-2859-10-S1-S20). We believe that LP+PR strain would be a potential candidate for industrial usage in animal production or fermentation.

4. This research refers to zoonotic pathogens however there is no such infos in literature except for human enteropathogens.

            Thank you for the comment. We have included briefly about zoonotic pathogens in the introduction section.

5. Lines 16-18: any recent ref?

            Lines 16-18 is in the abstract section. We have confusion about abstract citation or typing error from the reviewer.

6. In section 2.4 and 2.8 references should be provided

            Thank you for the suggestion. In accordant with the suggestion, we have provided the reference in section 2.4 and 2.8.

7. Section: 2.10 briefly describe the extraction method, did you use any phenol? Did you proceed to dnase treatment, measure RNA purity in some way?

We have briefly described the RNA extraction in the method section. We didn’t treat RNA with DNase and measured RNA concentration using NanoDrop. A 260/280 ratio of all the RNA samples were around 2.0.

8. Could you specify which method did you use for gene expression, Livak? Pfaffl? Cts were logged out? What is the expression ‘threshold’ of expression change?

            Based on the reviewer’s suggestion, we have included a brief description of qRT-PCR for evaluation of gene expressions in the method section with reference (Livak and Schmittgen, Analysis of relative gene expression data using real-time quantitative PCR and the 2-delta-delta-CT Method. Methods 2001, 25, 402–408. DOI: 10.1006/meth.2001.1262).

9. Discussion section: Not adequate with other studies outcomes.  Most of the literature provided is too old. If no comparison is feasible with other or current literature should be emphasized as it

            We thank the reviewer for giving this valuable comment. In response to the reviewer’s suggestion, we have changed older literature to recent literature in the discussion section.

Reviewer 3 Report

I strongly recommend that the author have the manuscript re-read by a native English speaker, as there are continual errors with pluralisation and with regards to the use of articles (a, an, the).   I would also suggest that the number of references be reduced slightly, as there are multiple references in the introduction that are possibly not all necessary.

Author Response

Reviewer #3:

1. I strongly recommend that the author have the manuscript re read by a native English speaker, as there are continual errors with pluralisation and with regards to the use of articles (a, an, the).

              We thank for the reviewer comment. Based on the reviewer’s suggestion, proofreading of this revised manuscript was done by native English speaker.

2. I would also suggest that the number of references be reduced slightly, as there are multiple references in the introduction that are possibly not all necessary.would be the novelty of the study.

            We thank the reviewer for the suggestion. As per the reviewer suggestion, we reduced the number of references in the revised manuscript.

Reviewer 4 Report

The study by Vinod Nagarajan et al., investigated the probiotic potential of phage-resistant Lactobacillus plantarum strain LP+PR, in comparison to phage-sensitive L. plantarum strain (LP). The authors demonstrated that LP+PR strain showed higher growth rate than LP strain. Moreover, LP+PR cells and the cell-free cultural supernatants of LP+PR inhibited the growth of pathogenic bacteria S. Typhimurium, EHEC, S. aureus, and L. monocytogenes. The authors also showed that LP+PR strain inhibited the adhesive and invasive abilities of all four pathogens in vitro. Moreover, real-time qPCR revealed that in the presence of LP+PR strain or its CFCSs, expression of virulence genes of these zoonotic bacterial pathogens were suppressed.

Although, there are several similar studies in the literature, and the adhesion properties of LP are well-described, the study by Vinod Nagarajan et al., is quite interesting. The objectives of the study are clear presented. The methodologies are described in detail. The results are clearly presented and the figures are well prepared.

I have the following minor comments:

1. The authors should present Figure 1 as supplementary data.  

2. The CFCSs from both strains showed strong growth inhibitory effects and they reduced both adherence and invasion abilities of all four pathogens. These are very promising results. Are the authors planning to study and characterize the metabolites of the CFCSs?

Author Response

Reviewer #4:

The study by Vinod Nagarajan et al., investigated the probiotic potential of phage-resistant Lactobacillus plantarum strain LP+PR, in comparison to phage-sensitive L. plantarum strain (LP). The authors demonstrated that LP+PR strain showed higher growth rate than LP strain. Moreover, LP+PR cells and the cell-free cultural supernatants of LP+PR inhibited the growth of pathogenic bacteria S. Typhimurium, EHEC, S. aureus, and L. monocytogenes. The authors also showed that LP+PR strain inhibited the adhesive and invasive abilities of all four pathogens in vitro. Moreover, real-time qPCR revealed that in the presence of LP+PR strain or its CFCSs, expression of virulence genes of these zoonotic bacterial pathogens were suppressed.

Although, there are several similar studies in the literature, and the adhesion properties of LP are well-described, the study by Vinod Nagarajan et al., is quite interesting. The objectives of the study are clear presented. The methodologies are described in detail. The results are clearly presented and the figures are well prepared.

I have the following minor comments:

1. The authors should present Figure 1 as supplementary data.  

            Thank you for the valuable suggestion. We are pleased to inform you that we have revised Figure 1 as supplementary data.

2. The CFCSs from both strains showed strong growth inhibitory effects and they reduced both adherence and invasion abilities of all four pathogens. These are very promising results. Are the authors planning to study and characterize the metabolites of the CFCSs?

            We thank the reviewer for the appreciative and insightful comment. Yes, we are planning to characterize the metabolites of the CFCSs in future studies.